# Emotion Classification Based on Transformer and CNN for EEG Spatial–Temporal Feature Learning

**DOI:** 10.3390/brainsci14030268

**Published:** 2024-03-11

**Authors:** Xiuzhen Yao, Tianwen Li, Peng Ding, Fan Wang, Lei Zhao, Anmin Gong, Wenya Nan, Yunfa Fu

**Affiliations:** 1Faculty of Information Engineering and Automation, Kunming University of Science and Technology, Kunming 650500, China; yaoxiuzhen22@gmail.com (X.Y.); ausarschorr@foxmail.com (P.D.); fanwang991@gmail.com (F.W.); 2Brain Cognition and Brain-Computer Intelligence Integration Group, Kunming University of Science and Technology, Kunming 650500, China; bushltw@hotmail.com (T.L.); ynzhaolei@foxmail.com (L.Z.); 3Faculty of Science, Kunming University of Science and Technology, Kunming 650500, China; 4School of Information Engineering, Chinese People’s Armed Police Force Engineering University, Xi’an 710086, China; gonganmincapf@foxmail.com; 5Department of Psychology, College of Education, Shanghai Normal University, Shanghai 200234, China; wynan1985@126.com

**Keywords:** EEG, emotion classification, transformer, CNN, multi-head attention

## Abstract

Objectives: The temporal and spatial information of electroencephalogram (EEG) signals is crucial for recognizing features in emotion classification models, but it excessively relies on manual feature extraction. The transformer model has the capability of performing automatic feature extraction; however, its potential has not been fully explored in the classification of emotion-related EEG signals. To address these challenges, the present study proposes a novel model based on transformer and convolutional neural networks (TCNN) for EEG spatial–temporal (EEG ST) feature learning to automatic emotion classification. Methods: The proposed EEG ST-TCNN model utilizes position encoding (PE) and multi-head attention to perceive channel positions and timing information in EEG signals. Two parallel transformer encoders in the model are used to extract spatial and temporal features from emotion-related EEG signals, and a CNN is used to aggregate the EEG’s spatial and temporal features, which are subsequently classified using Softmax. Results: The proposed EEG ST-TCNN model achieved an accuracy of 96.67% on the SEED dataset and accuracies of 95.73%, 96.95%, and 96.34% for the arousal–valence, arousal, and valence dimensions, respectively, for the DEAP dataset. Conclusions: The results demonstrate the effectiveness of the proposed ST-TCNN model, with superior performance in emotion classification compared to recent relevant studies. Significance: The proposed EEG ST-TCNN model has the potential to be used for EEG-based automatic emotion recognition.

## 1. Introduction

Emotion is a comprehensive representation of an individual’s subjective experience and behavior, encompassing various feelings, behaviors, and thoughts. It significantly influences an individual’s perception and attitude, and its identification has potential applications [1]. There are two primary types of methods used for emotion recognition [2]. One involves external responses evoked by emotions, such as facial expressions and gestures [3], while the other focuses on internal responses induced by emotions like electroencephalogram (EEG) and electrocardiogram signals, along with other physiological signals [4]. Compared with non-physiological signals, physiological signals are less easily controlled subjectively by individuals and are challenging to conceal [5]. Among the several physiological signals, EEG is an electrical signal generated by the central nervous system. Nunez et al. discovered that EEG signals exhibit certain associations with physiological events such as sleep patterns, neurological disorders, and emotional states [6]. Existing studies have been conducted regarding the recognition of individual emotional states using EEG [7], suggesting that EEG holds promise as a physiological signal for emotion recognition.

The current methods for emotion recognition based on EEG include two primary methods: traditional machine learning and deep learning. Bhardwaj et al. [8] classified seven different emotions based on EEG signals, preprocessed the data using filtering and an independent component analysis (ICA), and then extracted the energy and power spectral density (PSD) as features. The average classification accuracies achieved using a support vector machine (SVM) and linear discriminant analysis (LDA) were 74.13% and 66.50%, respectively. Wang et al. [9] utilized the minimum redundancy maximum relevance method to extract key EEG features for four types of emotions and compared its classification accuracy to that of of k-nearest neighbors (KNN), multi-layer perceptron, and SVM, demonstrating that the frequency domain features combined with SVM achieved an average accuracy of 66.51%. Xiao et al. [10] proposed a 4D attention neural network that involved the conversion of the raw EEG into a spatial–spectral–temporal representation in four dimensions. Subsequently, a CNN was used to process both spectral and spatial information, while attention mechanisms were integrated into a bidirectional long short-term memory network (Bi-LSTM) for processing temporal information. The deep learning model achieved an average accuracy of 96.90% and 97.39% in the valence and arousal dimensions, respectively, for the DEAP dataset [11]. Additionally, it achieved an average accuracy of 96.25% for the SEED dataset [12], encompassing three types of emotions: positive, neutral, and negative. Furthermore, it achieved an accuracy of 86.77% for the SEED-IV dataset [13], which includes four types of emotions: happy, sad, fear, and neutral.

The proposed EEG emotion recognition algorithm by An et al. [13] is based on 3D feature fusion and a convolutional autoencoder (CAE). First, the differential entropy (DE) features from various frequency bands of EEG were fused to construct 3D features of EEG signals, which preserve spatial information between channels. Then, the constructed 3D features were input into the CAE for emotion recognition. The deep learning model achieved an average accuracy of 89.49% for the valence dimension and 90.76% for the arousal dimension when evaluated on the DEAP dataset.

The aforementioned studies demonstrate the effectiveness of traditional machine learning and deep learning techniques in EEG-based emotion classification involving the extraction and selection of an optimal feature set from raw EEG data. However, the process of extracting features from raw EEG signals may result in the loss of valuable information, thereby hindering the model’s ability to learn missing information. In this study, transformers are utilized to automatically extract the spatial–temporal features that are relevant to emotions from EEG data, aiming to mitigate the loss of valuable information. Additionally, a CNN is used for aggregating these extracted spatial–temporal features as a means to address the aforementioned challenges.

The transformer model, initially proposed by Vaswani et al. in 2017 [14], has demonstrated remarkable success in the fields of natural language processing [15] and computer vision [16]. Unlike CNN, RNN, and LSTM networks, this model overcomes the limitations of local receptive fields and enables concurrent consideration of information across all positions in a sequence. Consequently, it excels at capturing global relationships and exhibits superior performance in calculating correlations among features within long sequences while effectively processing long-term dependencies. In this study, we anticipate that leveraging the power of the transformer model will enable effective extraction of the spatial–temporal features of emotion-related EEG signals, thereby improving the accuracy of emotion classification.

## 2. Materials and Methods

### 2.1. Datasets

The SEED dataset, publicly released by Shanghai Jiao Tong University, is widely utilized in the fields of emotion recognition and emotion-related BCI research [13]. The experimental approach involves using movie clips as stimuli to evoke specific emotions, including positive, neutral, and negative emotions. Fifteen participants (seven males, eight females) with an average age of 23.27 were involved in the experiment. Each session required the participants to watch a set of 15 movie clips, each lasting approximately 4 min. These clips covered positive, neutral, and negative emotions, with five clips corresponding to each emotional category [17]. The EEG recordings were obtained using the ESI NeuroScan system, with a 62-channel electrode distribution following the international 10–20 standard system. The EEG channels included FP1, FPZ, FP2, AF3, AF4, F7, F5, F3, F1, FZ, F2, F4, F6, F8, FT7, FC5, FC3, FC1, FCZ, FC2, FC4, FC6, FT8, T7, C5, C3, C1, CZ, C2, C4, C6, T8, TP7, CP5, CP3, CP1, CPZ, CP2, CP4, CP6, TP8, P7, P5, P3, P1, PZ, P2, P4, P6, P8, PO7, PO5, PO3, POZ, PO4, PO6, PO8, CB1, O1, OZ, O2, and CB2. The data were initially sampled at a rate of 1000 Hz and subsequently downsampled to 200 Hz. Furthermore, filtering was performed using a bandpass filter with a frequency range of 0–75 Hz.

The DEAP dataset, established by Sanders Koelstra et al. [12], is a publicly available multimodal emotion dataset focusing on emotional dimensions. The dataset consists of 32-channel EEG recordings from 32 participants while they watched 40 one-minute-long music video clips. After watching each clip, the participants were asked to self-rate their arousal and valence levels on a scale ranging from 1 to 9. The EEG channels included Fp1, AF3, F3, F7, FC5, FC1, C3, T7, CP5, CP1, P3, P7, PO3, O1, Oz, Pz, Fp2, AF4, Fz, F4, F8, FC6, FC2, Cz, C4, T8, CP6, CP2, P4, P8, PO4, and O2. To enhance the data quality, the raw EEG signals were downsampled to a frequency of 128 Hz and filtered using a bandpass filter with cutoff frequencies of 4 Hz and 45 Hz.

### 2.2. Preprocessing

The study conducted by Lashgari et al. [18] demonstrates that segmenting and reassembling data in the time domain preserves information, enables data expansion, and enhances classification accuracy. Previous research has demonstrated that implementing a 3 s time window leads to improved classification accuracy [19]. Therefore, this study employs a 3 s time window for EEG segmentation.

### 2.3. Experimental Platform

The entire experimental process was conducted in an environment with an Intel(R) Core i5-12400F processor and an NVIDIA RTX3080 Ti GPU. The model was implemented using the Python 3.7 programming language and the Pytorch deep learning framework.

### 2.4. Experimental Procedures

The flowchart in Figure 1 illustrates the procedures of emotion classification using the transformer and a CNN based on EEG spatial and temporal feature learning. Initially, the raw EEG signals are preprocessed and segmented, and then emotion-related EEG spatial and temporal features are extracted by the transformer-based module. Finally, the results are generated by a prediction layer including a CNN, maximum pooling (MaxPooling), fully connected (FC), and Softmax layer.

### 2.5. Transformer Encoder

The encoder of the transformer model used in this study is illustrated in Figure 2. The transformer encoder includes scaled dot-product attention, addition and normalization (Add and Norm), multi-head attention, and a feed-forward neural network. The scaled dot-product attention mechanism in the transformer encoder is closely connected to the multi-head attention mechanism. In the scaled dot-product attention, the input sequence is linearly transformed to obtain query, key, and value vectors. A scaling dot product operation is then performed to calculate the attention score, which adjusts the dot product value by dividing it by the square root of the dimension of the query and key vectors. This ensures gradient stability. On the other hand, because multiple heads perform independent scaling dot product operations, the multi-head attention mechanism enables parallel focus on various aspects of the input sequence. By concatenating and finally linearly transforming these head outputs, a comprehensive representation is obtained for each position. Through this tight correlation between the scaled dot-product attention and multi-head attention, efficient encoding and representation of the input sequences are achieved by the transformer encoder.

#### 2.5.1. Scaled Dot-Product Attention

The scaled dot-product attention is depicted in Figure 2a. Initially, the input data are multiplied by three different weight matrices to obtain the query vector (Q), key vector (K) and value vector (V), respectively. Subsequently, the dot product of Q and K is divided by the scaling factor $\sqrt{d_{k}}$ (where $d_{k}$ represents the dimension of the query vector), followed by computation of the weights using a Softmax function. Finally, these weights are multiplied by the value vector V to obtain a weighted result. The calculation is performed using Equation (1).
(1)$$\mathrm{Attention}(Q,K,V)=\mathrm{Softmax}(\frac{{\mathrm{QK}}^{T}}{\sqrt{d_{k}}})V$$

#### 2.5.2. Multi-Head Attention

In Figure 2b, the multi-head attention mechanism consists of h scaled dot-product attention layers. Each scaled dot-product attention layer focuses on the information found in different subspaces. This structure allows the model to concurrently process various aspects of correlations, thereby comprehensively capturing the features of the input data. Subsequently, the attention representations from these diverse heads are concatenated to form the final multi-head attention representation, calculated using Equation (2), where $W^{O}$ is the weight matrix of the output.
(2)$${\mathrm{MultiHead}(Q,K,V)=\mathrm{Concat}(\mathrm{head}}_{1},\ldots {,\mathrm{head}}_{h}{)W}^{O}$$

#### 2.5.3. Transformer Encoder

The transformer encoder in Figure 2c is mainly composed of two modules. The first module includes a multi-head attention layer and a normalization layer, with the latter being used to improve the stability and accelerate convergence during training. Additionally, residual connections are used between these layers to facilitate information flow and alleviate the issue of gradient vanishing. The second module is composed of a feedforward neural network layer and a normalization layer. The feedforward neural network nonlinearly maps the features obtained from the multi-head attention mechanism, contributing to the model’s ability to capture distinctive features within the input sequence. Residual connections are also utilized between these layers. This architectural design aims to fully leverage the self-attention mechanism and feedforward network in the transformer model to effectively capture contextual information from the EEG signals and improve classification performance.

### 2.6. Transformer and CNN Models for Learning Emotion-Related EEG Temporal and Spatial Features

This study proposes a novel emotion classification model, named EEG spatial–temporal transformer and CNN (EEG ST-TCNN), which is based on EEG spatial–temporal feature learning, as illustrated in Figure 3. The model effectively uses both the spatial and temporal information embedded in EEG signals.

The EEG ST-TCNN model consists of three components. The first component is the input module of the model, where the raw EEG signals are concurrently input into the model in both spatial and temporal arrangements. The batch_size, channel, and time_point were set to 128, 62, and 600, respectively, during the experimentation on the SEED dataset. Similarly, for the DEAP dataset, the batch_size was set to 128, while the channel and time_point were adjusted to 32 and 384, respectively. To capture the EEG sequence information effectively, positional encoding (PE) is applied to embed the input EEG. PE plays a crucial role in understanding the relationship between element position and order when processing sequence data with transformers, thereby enhancing their performance in various natural language processing and sequence modeling tasks [14]. In this study, we adopt the method of relative positional encoding, which is implemented using Equations (3) and (4):(3)PE(pos, 2i)=sin(pos100002id)
(4)PE(pos, 2i+1)=cos(pos100002id)
where PE(pos, 2i) and PE(pos, 2i + 1) denote the two elements of the positional encoding at position pos and dimension i, respectively. Here, d denotes the dimension of the embedding vector.

In the second component, two transformer encoders are used to extract the deep spatial and temporal features from the input EEG. The third component consists of a prediction layer, which consists of 2 convolution layers (with a convolution kernel size of 3 × 3, and there are 64 convolutional kernels), 1 MaxPooling layer (with a window size of 2 × 2), 1 FC layer, and 1 Softmax layer. Within the prediction layer, the spatial and temporal features extracted by the transformer encoder are concatenated. This concatenated representation is then processed by a combination of CNN and MaxPooling layers to effectively aggregate the features and capture the local features. Subsequently, the processed features are passed into the FC before being classified into different emotional states by Softmax.

### 2.7. Model Training Strategy and Process

In order to train and validate the proposed model, a ten-fold cross-validation method was used to divide the EEG data of each subject into ten samples, of which nine were used as training samples and one was used as a test sample. Finally, the average accuracy was calculated as the classification result. In the multi-head attention mechanism of the transformer, the parameter “h” was set to 8, and cross-entropy along with L2 regularization terms were utilized as the loss functions. The Adam optimizer was used for optimization. During the training process, the parameters learning rate and “batch_size” were set to 0.0001 and 128, respectively. Additionally, a dropout value of 0.3 was set to prevent overfitting, and the ReLU function was used as the activation function.

### 2.8. Evaluation Metrics

The classification performance of the model is evaluated using accuracy (Acc), precision (P), recall (R), and F1-score as evaluation metrics, which are calculated using Equations (5)–(8):(5)$$\mathrm{Acc}=\frac{\mathrm{TP}+\mathrm{TN}}{\mathrm{TP}+\mathrm{TN}+\mathrm{FP}+\mathrm{FN}}$$
(6)$$P=\frac{\mathrm{TP}}{\mathrm{TP}+\mathrm{FP}}$$
(7)$$R=\frac{\mathrm{TP}}{\mathrm{TP}+\mathrm{FN}}$$
(8)$$F1\;\mathrm{Score}=2\times \frac{P\times R}{P+R}$$
where TP indicates that the prediction of the positive class is positive, TN indicates that the prediction of the negative class is negative, FP indicates that the prediction of the negative class is positive, and FN indicates that the prediction of the positive class is negative.

## 3. Results

### 3.1. Classification Performance

We conducted independent experiments on both the SEED and DEAP datasets. Four models were systematically obtained by progressively removing a certain module from the model, including EEG ST-TCNN, the model without CNN (EEG ST-T), the model with only spatial dimensions as input (EEG S-T), and the model with only temporal dimensions as input (EEG T-T). The classification results of these four models were compared to validate the effectiveness of the removed modules in the model. Figure 4 shows the accuracy of emotion recognition for these aforementioned models. Both the EEG ST-TCNN and EEG ST-T models achieved high accuracy on the SEED and DEAP datasets. Compared to the EEG ST-T model, the EEG ST-TCNN model demonstrated improvements of 0.69%, 1.93%, 0.78%, and 1.9% in the positive–neutral–negative, arousal–valence, arousal, and valence dimensions, respectively. The average accuracy and variance of the four models in the different dimensions are presented in Table 1, while Table 2 displays the *t*-test results for the accuracies of these models across the different dimensions. The results of EEG ST-TCNN and EEG ST-T did not show significant differences in the arousal dimension, whereas there were significant differences between the results of EEG ST-TCNN and other models in the positive–neutral–negative, arousal–valence, and valence dimensions.

Table 3, Table 4, Table 5 and Table 6 present the accuracy, precision, recall, and F1-score achieved in experiments on the SEED and DEAP datasets for the EEG ST-TCNN, EEG ST-T, EEG S-T, and EEG T-T models. The experimental results demonstrate that utilizing the raw EEG spatial and temporal arrangement input concurrently is superior to solely using the spatial or temporal arrangements as inputs. Compared with EEG ST-T, the integration of the CNN into the EEG ST-T model leads to enhanced effects across the various dimensions.

The confusion matrices in Figure 5, Figure 6, Figure 7 and Figure 8 illustrate the results obtained by the four different models across the positive–neutral–negative, arousal–valence, arousal, and valence dimensions. In each figure, (a) represents the confusion matrix achieved by EEG ST-TCNN in the corresponding dimension experiment; (b) represents the confusion matrix achieved by EEG ST-T in the corresponding dimension experiment; (c) represents the confusion matrix achieved by EEG S-T in the corresponding dimension experiment; and (d) represents the confusion matrix achieved by EEG T-T in the corresponding dimension experiment. The experimental results demonstrate that the proposed EEG ST-TCNN shows superior performance on both the SEED and DEAP datasets.

### 3.2. Comparison of the Results Obtained Using Other Methods

Shen et al. [2] firstly calculated the differential entropy (DE) features from different EEG channels signals and then converted them into a four-dimensional structure. The structure integrated the frequency domain, time domain, and spatial domain features of the EEG signals. Subsequently, these structured data were then input into a four-dimensional convolutional recurrent neural network (4D-CRNN) for training. Liu et al. [20] constructed an undirected graph based on the spatial relationships between EEG electrodes. They utilized the differential entropy features of the EEG signals to represent the nodes of the undirected graph. Furthermore, they proposed a model for emotion recognition based on EEG signals using a global-to-local feature aggregation network (GLFANet). Finally, the undirected graph was fed into this model. Zheng et al. [21] constructed an EEG electrode location matrix corresponding to brain region distribution, thereby reconstructing EEG data. They used a combined model of a graph convolutional neural network and LSTM (GCN + LSTM) to extract the spatial and temporal features of the EEG signals. Liu et al. [22] proposed a model that combines a convolutional neural network (CNN), sparse autoencoder, and deep neural network (CNN-SAE-DNN). They integrated the frequency–domain features and spatial location information of EEG signals to construct a two-dimensional data input for the model. Yang et al. [23] utilized various frequency–domain features of EEG signals to construct three-dimensional data, which was then fed into a continuous convolutional neural network (continuous CNN). Table 7 compares the classification performance of these classical or cutting-edge deep learning methods with the proposed EEG ST-TCNN model on the SEED and DEAP datasets. The results demonstrate that the proposed model exhibits varying degrees of improvement in classification performance on both the SEED and DEAP datasets.

## 4. Discussion

To address the prevalent challenges in current EEG-based emotion recognition, which frequently rely on manual feature extraction and the selection of an optimal feature set, this paper proposes a novel EEG ST-TCNN model.

The robustness of the proposed model was validated through independent experiments conducted on two publicly available datasets in this study. To validate the effectiveness of each module in the model, we progressively removed one module and compared the experimental results. As illustrated in Figure 4 and Table 3, Table 4, Table 5 and Table 6, the ST model exhibits inferior performance compared to the EEG T-T model overall. However, the EEG S-T model contributes to enhancing the spatial location information. Notably, concurrently utilizing a spatial and temporal arrangement input based on raw EEG data in the experiment is superior to using the single spatial or temporal arrangement input alone. Both EEG ST-TCNN and EEG ST-T demonstrate superior suitability for emotion classification based on EEG signals. Compared with EEG ST-T, EEG ST-TCNN exhibits improved performances in different dimensions. Importantly, EEG ST-TCNN excels in integrating spatial and temporal information from the raw EEG signals, thereby enhancing the accuracy of emotion recognition.

In this study, the temporal and spatial information from the raw EEG signals is taken into consideration, while its frequency domain features are not utilized. Relative positional encoding is used in this research; however, the impact of different positional encoding methods on emotion classification based on EEG signals remains unexplored. It is noteworthy that Wu et al. [24] demonstrated that various positional encoding methods can affect transformer performance. Furthermore, individual differences among subjects were not taken into account in this study. Importantly, other physiological signals also contribute to the task of emotion recognition. Sun et al. [25] proposed a bimodal method combining functional near-infrared spectroscopy (fNIRS) and EEG to identify emotions, and their results indicated superior performance of the fNIRS+EEG method compared to using only fNIRS or EEG.

In our future research, we will combine frequency domain features with the spatial and temporal information of raw EEG signals and investigate the impact of different position encoding methods on model classification results prior to integrating EEG temporal, spatial, and frequency features into the model. Furthermore, we intend to conduct multi-modal and cross-subject emotion recognition research, with the expectation that the proposed model will further enhance the performance of emotion classification based on EEG signals. Compared to offline emotion recognition, we believe that real-time emotion classification based on EEG holds greater significance and potential applications. Real-time emotion recognition can provide individuals with immediate feedback, thus enhancing the effectiveness of emotion regulation. In future work, we will pay more attention to the complexity of algorithms and endeavor to classify emotions in real-time based on EEG signals.

## 5. Conclusions

The present study proposes a novel EEG STT-CNN model that is designed to address the challenge of emotion classification based on EEG signals. Specifically, two parallel transformer encoders are used to extract deep spatial and temporal features from emotion-related EEG data. Subsequently, these features are integrated using a CNN. The experimental results demonstrate that the proposed model achieves an accuracy of 96.67% on the SEED dataset. Additionally, it attains accuracies of 95.73%, 96.95%, and 96.34% on the DEAP dataset for the arousal–valence, arousal, and valence dimensions, respectively. A comparative analysis with recent relevant research indicates that the proposed model outperforms existing methods, highlighting its potential for automated EEG-based emotion recognition.

## Figures and Tables

**Figure 1 brainsci-14-00268-f001:**
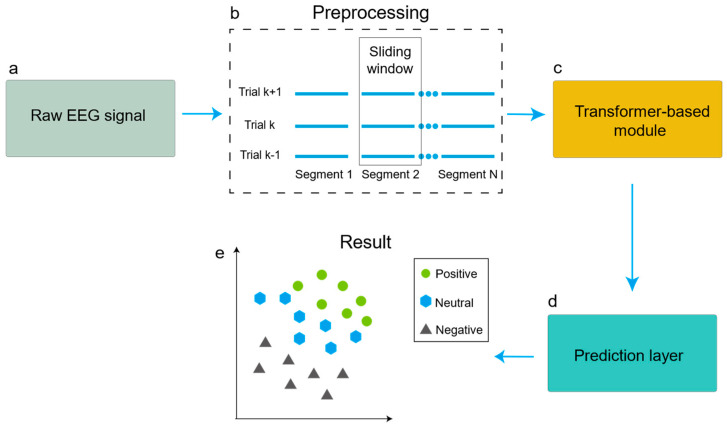
The flowchart of emotion classification using the transformer and a CNN based on EEG spatial–temporal feature learning. (**a**) Acquisition of input data; (**b**) data preprocessing steps, during which EEG signals are segmented; (**c**) the transformer-based module, which extracts the spatial and temporal features of the EEG signals; (**d**) the prediction layer, which aggregates the extracted temporal and spatial features and performs the classification; (**e**) the classification results.

**Figure 2 brainsci-14-00268-f002:**
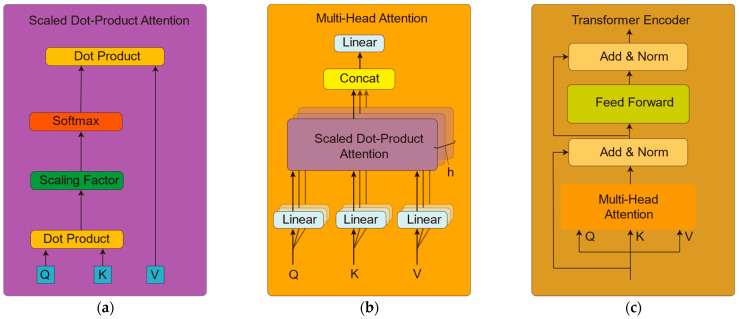
The structure of the transformer encoder. (**a**) scaled dot-product attention; (**b**) multi-head attention; (**c**) transformer encoder.

**Figure 3 brainsci-14-00268-f003:**
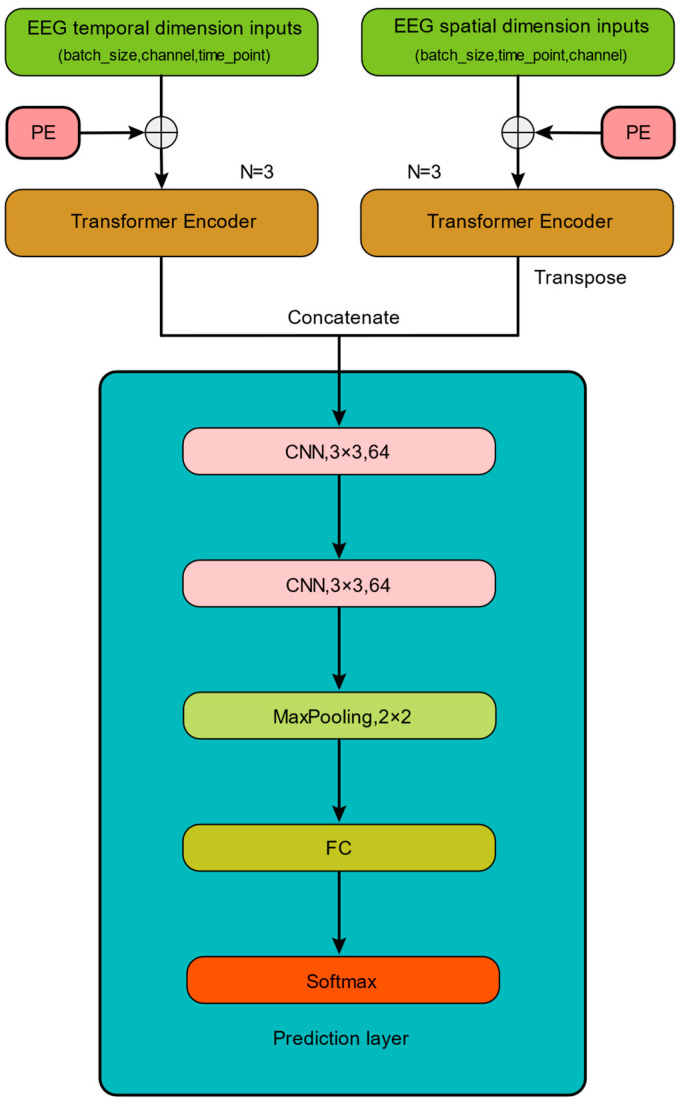
The EEG ST-TCNN model. Note: PE (positional encoding); (batch_size, channel, time_point) and (batch_size, time_point, channel) represent the temporal and spatial arrangements of the EEG signals, respectively.

**Figure 4 brainsci-14-00268-f004:**
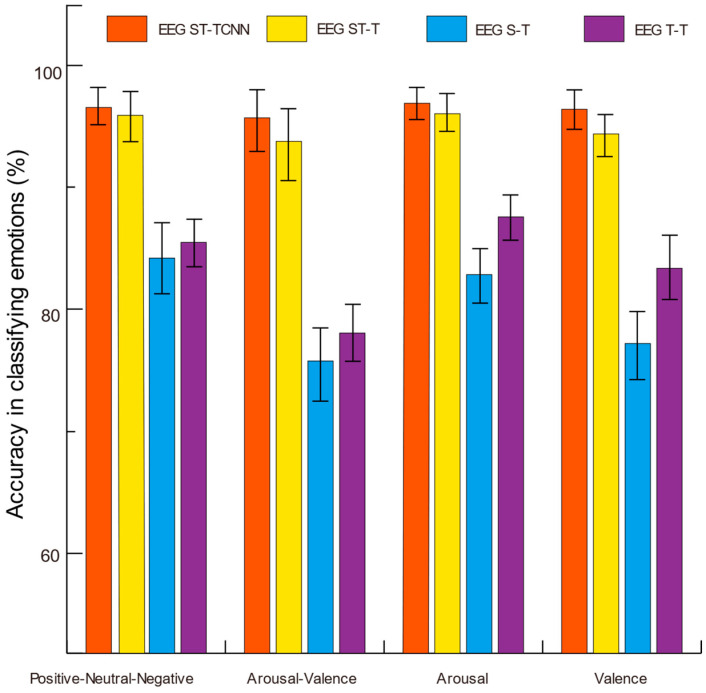
The accuracy of the four models in classifying emotions on the SEED and DEAP datasets.

**Figure 5 brainsci-14-00268-f005:**
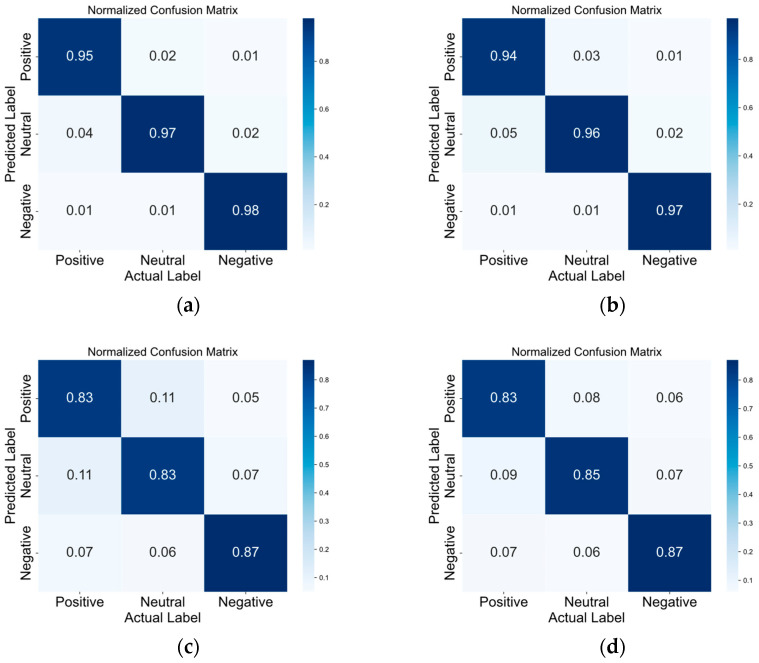
The confusion matrices of the four models in the positive–neutral–negative dimension. Note: (**a**–**d**) respectively represent the results obtained by EEG ST-TCNN, EEG ST-T, EEG S-T, and EEG T-T.

**Figure 6 brainsci-14-00268-f006:**
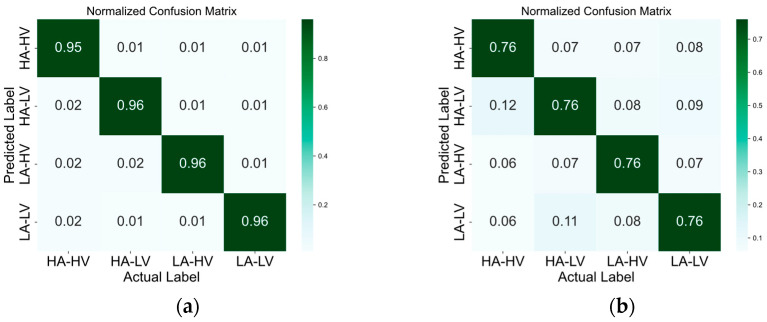
The confusion matrices of the four models in the arousal–valence dimension. Note: (**a**–**d**) respectively represent the results obtained by EEG ST-TCNN, EEG ST-T, EEG S-T, and EEG T-T.

**Figure 7 brainsci-14-00268-f007:**
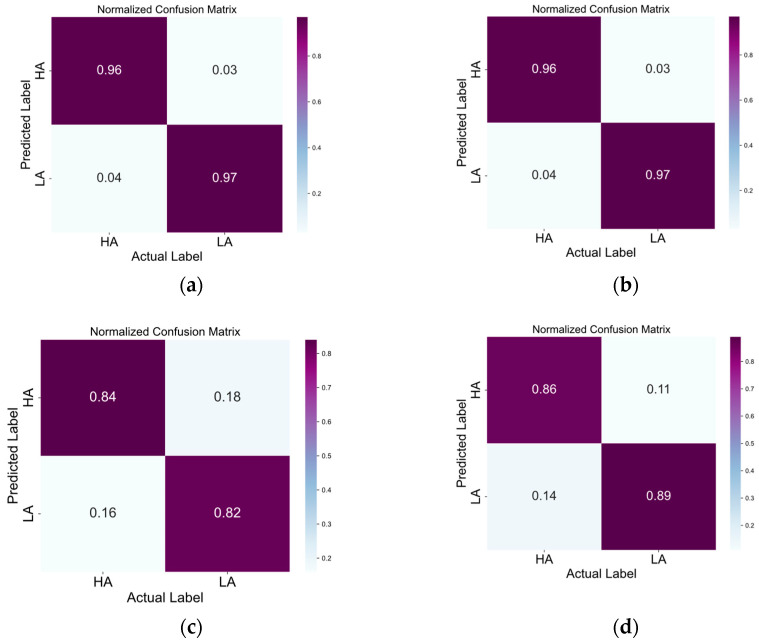
The confusion matrices of the four models in the arousal dimension. Note: (**a**–**d**) respectively represent the results obtained by EEG ST-TCNN, EEG ST-T, EEG S-T, and EEG T-T.

**Figure 8 brainsci-14-00268-f008:**
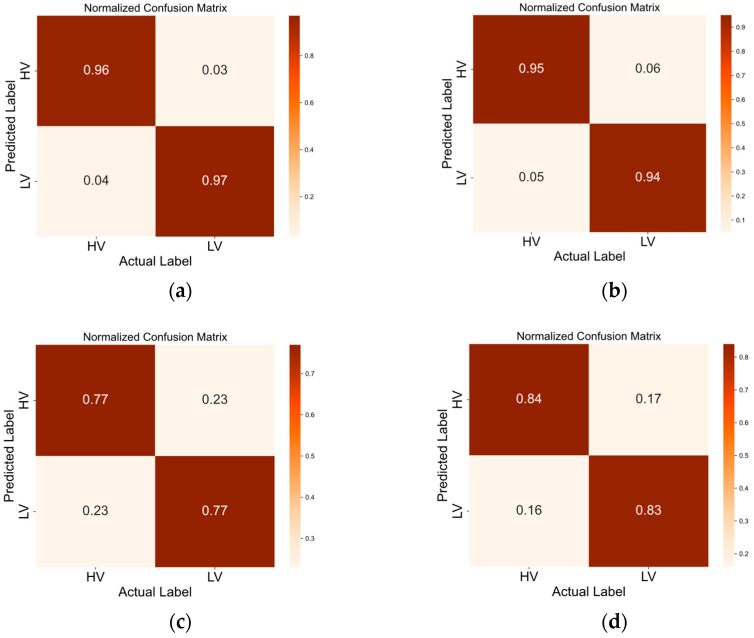
The confusion matrices of the four models in the valence dimension. Note: (**a**–**d**) respectively represent the results obtained by EEG ST-TCNN, EEG ST-T, EEG S-T, and EEG T-T.

**Table 1 brainsci-14-00268-t001:** The average accuracy and variance of the four models across the different dimensions.

Dimension	Model	Acc/%	Variance
Positive–neutral–negative	EEG ST-TCNN	96.67	2.88
	EEG ST-T	95.98	3.11
	EEG S-T	84.29	5.66
	EEG T-T	85.27	3.71
Arousal–valence	EEG ST-TCNN	95.73	5.40
	EEG ST-T	93.80	6.19
	EEG S-T	75.97	5.66
	EEG T-T	78.03	6.35
Arousal	EEG ST-TCNN	96.95	2.04
	EEG ST-T	96.17	3.42
	EEG S-T	82.93	4.27
	EEG T-T	87.65	2.25
Valence	EEG ST-TCNN	96.34	3.02
	EEG ST-T	94.44	3.46
	EEG S-T	77.22	5.26
	EEG T-T	83.40	5.00

**Table 2 brainsci-14-00268-t002:** *t*-test results: Comparing the accuracies across the different dimensions for the four models.

Dimension	Comparison	*t*-Value	*p*-Value
Positive–neutral–negative	EEG ST-TCNN vs. EEG ST-T	2.07	0.045
	EEG ST-TCNN vs. EEG S-T	17.54	<0.001
	EEG ST-TCNN vs. EEG T-T	17.19	<0.001
	EEG ST-T vs. EEG S-T	16.47	<0.001
	EEG ST-T vs. EEG T-T	15.87	<0.001
	EEG S-T vs. EEG T-T	−1.22	0.232
Arousal–valence	EEG ST-TCNN vs. EEG ST-T	2.18	0.036
	EEG ST-TCNN vs. EEG S-T	21.22	<0.001
	EEG ST-TCNN vs. EEG T-T	19.56	<0.001
	EEG ST-T vs. EEG S-T	20.65	<0.001
	EEG ST-T vs. EEG T-T	18.89	<0.001
	EEG S-T vs. EEG T-T	−2.03	0.049
Arousal	EEG ST-TCNN vs. EEG ST-T	2.01	0.053
	EEG ST-TCNN vs. EEG S-T	17.51	<0.001
	EEG ST-TCNN vs. EEG T-T	17.13	<0.001
	EEG ST-T vs. EEG S-T	17.67	<0.001
	EEG ST-T vs. EEG T-T	14.18	<0.001
	EEG S-T vs. EEG T-T	−5.84	<0.001
Valence	EEG ST-TCNN vs. EEG ST-T	2.09	0.044
	EEG ST-TCNN vs. EEG S-T	22.20	<0.001
	EEG ST-TCNN vs. EEG T-T	17.66	<0.001
	EEG ST-T vs. EEG S-T	19.65	<0.001
	EEG ST-T vs. EEG T-T	15.42	<0.001
	EEG S-T vs. EEG T-T	−7.28	<0.001

**Table 3 brainsci-14-00268-t003:** The accuracy, precision, recall, and F1-score of the four models in the positive–neutral–negative dimension.

Model	Acc/%	P/%	R/%	F1-Score/%
EEG ST-TCNN	96.67	96.70	96.64	96.67
EEG ST-T	95.98	95.64	95.54	95.59
EEG S-T	84.29	84.26	84.25	84.25
EEG T-T	85.27	85.27	85.24	85.25

**Table 4 brainsci-14-00268-t004:** The accuracy, precision, recall, and F1-score of the four models in the arousal–valence dimension.

Method	Acc/%	P/%	R/%	F1-Score/%
EEG ST-TCNN	95.73	95.72	95.72	95.72
EEG ST-T	93.80	93.80	93.78	93.79
EEG S-T	75.97	76.07	75.97	76.02
EEG T-T	78.03	78.15	78.03	78.09

**Table 5 brainsci-14-00268-t005:** The accuracy, precision, recall, and F1-Score of the four models in the arousal dimension.

Method	Acc/%	P/%	R/%	F1-Score/%
EEG ST-TCNN	96.95	96.95	96.91	96.95
EEG ST-T	96.17	96.17	96.17	96.17
EEG S-T	82.93	82.94	82.93	82.94
EEG T-T	87.65	87.69	87.65	87.67

**Table 6 brainsci-14-00268-t006:** The accuracy, precision, recall, and F1-score of the four models in the valence dimension.

Method	Acc/%	P/%	R/%	F1-Score/%
EEG ST-TCNN	96.34	96.36	96.33	96.34
EEG ST-T	94.44	94.44	94.45	94.44
EEG S-T	77.22	77.21	77.22	77.22
EEG T-T	83.40	83.40	83.41	83.41

**Table 7 brainsci-14-00268-t007:** Classification performance of some classic or state-of-the-art deep learning methods compared with that of the proposed model on the SEED and DEAP datasets. P-N-N represents positive–neutral–negative, A stands for arousal, and V denotes valence.

References	Methods	Features	P-N-N	A-V	A	V
Shen et al. [2]	4D-CRNN	DE	94.74	-	94.58	94.22
Liu et al. [20]	GLFANet	DE	-	92.92	94.91	94.53
Zheng et al. [21]	CNN-SAE-DNN	DE	94.92	-	90.33	90.67
Liu et al. [22]	GCN + LSTM	spatial–temporal features	96.77	-	92.86	89.49
Yang et al. [23]	Continuous CNN	DE	-	85.30	90.24	89.45
Ours	EEG ST-TCNN	Raw EEG	96.67	95.73	96.95	96.34

## Data Availability

The data that support the findings of this study are available from the SEED (https://bcmi.sjtu.edu.cn/~seed/downloads.html#seed-access-anchor (accessed on 1 December 2023)) and DEAP (https://www.eecs.qmul.ac.uk/mmv/datasets/deap/download.html (accessed on 1 December 2023)) public datasets.

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
