# Peer review of "Emotion Classification Based on Transformer and CNN for EEG Spatial–Temporal Feature Learning"

_brainsci, 2024, doi:10.3390/brainsci14030268_

Round 1

Reviewer 1 Report

Comments and Suggestions for Authors

The author has to clarify why the temporal and spatial components of the EEG were combined. CNN's architecture is not sufficient for understanding. Preprocessing techniques have not received enough attention. Emotions other than positive, negative, and neutral—such as anger, disgust, fear, happiness, sadness, and surprise—should be the researcher's primary focus. considering that the info is obtained from watching films. The research suggests using LSTM to implement data sequencing (Page 6).

Author Response

Dear Reviewer, 
Thank you for reviewing our paper and providing valuable feedback. We greatly appreciate your attention to our research and are willing to provide detailed responses and revisions to the questions you have raised.

Below are the corresponding revisions and responses:

(1)    Since the temporal and spatial components of the EEG are critical for recognizing emotions, we employ two parallel Transformer Encoders to process the temporal and spatial components, enabling the model to focus on the features of these two aspects separately and effectively combine them through subsequent fusion layers. The temporal and spatial features extracted from Transformer Encoder are independent, so we use the structure of CNN. Through the experimental results, it is found that the combination of temporal and spatial components has the best effect in this experiment.
(2)    Using preprocessing techniques can improve the classification performance of most models. In this study, the purpose is to classify raw EEG data, so only segmentation processing was performed.
(3)    Your suggestion is excellent, but this study focuses on the SEED and DEAP public datasets. The SEED dataset categorizes emotions into positive, negative, and neutral emotions, while the DEAP dataset categorizes emotions from the perspective of arousal and valence. In future plans, we will focus on anger, disgust, fear, happiness, sadness, and surprise. Given your suggestion, we have included this plan in the discussion section.
(4)    While LSTM can indeed be used for sequential data, this study utilizes a Transformer Encoder. Transformers exhibit superior long-range dependency modeling, which typically leads to better performance and results when processing sequential data. Leveraging the feature extraction capability of CNNs allows for better fusion of spatial-temporal concatenations.

Thank you again for your valuable feedback.

Reviewer 2 Report

Comments and Suggestions for Authors

Contributions: The study proposes a novel model called ST-TCNN that combines Transformer and Convolutional Neural Networks (CNN) for EEG spatial-temporal (EEG ST) feature learning to automatically classify emotions. The model utilizes Position Encoding (PE) and multi-head attention to perceive channel positions and timing information in EEG signals. Two parallel Transformer Encoders in the model are used to extract spatial and temporal features from emotion-related EEG data.

Comments/Suggestions:

  • Provide more details on the SEED and DEAP datasets used in the study to enhance reproducibility and transparency.
  • Include a comparison of the proposed ST-TCNN model with other state-of-the-art emotion classification models to further validate its superiority.
  • Conduct additional experiments on different datasets to evaluate the generalizability of the proposed model.
  • Discuss the limitations of the proposed model and potential areas for future research to provide a comprehensive understanding of the study's scope.
  • Include a detailed explanation of the multi-head attention mechanism in the Transformer model and its relevance to emotion classification.
  • Provide a clear explanation of the Position Encoding (PE) technique used in the model and its impact on capturing spatial and temporal information in EEG signals.
  • Discuss the computational complexity and efficiency of the proposed model compared to other existing models to assess its practicality for real-time emotion recognition applications.
  • Consider conducting user studies or subjective evaluations to assess the perceived emotional accuracy of the proposed model and its potential applications in real-world scenarios.
  • The authors are invited to include some recent references, especially those related to  Deep Convolutional Neural Networks.
  • For instance, the authors may include the following interesting references (and others):

    a. https://www.mdpi.com/2073-431X/12/8/151

    b. https://www.taylorfrancis.com/chapters/edit/10.1201/9781003393030-10/learning-modeling-technique-convolution-neural-networks-online-education-fahad-alahmari-arshi-naim-hamed-alqa
Comments on the Quality of English Language

Can be improved.

Author Response

Dear Reviewer, 
Thank you for reviewing our paper and providing valuable feedback. We greatly appreciate your attention to our research and are willing to provide detailed responses and revisions to the questions you have raised.

Below are the corresponding revisions and responses:

In the introduction of the dataset, we supplemented which channels were used for each of these two datasets.

We conducted additional experiments on different datasets, and in future work, we plan to collect data.

In the discussion section, we have already explored the limitations of the proposed model, and following your suggestion, we have added potential areas for future research in the discussion section.

In the introduction of the Transformer Encoder, we provided a more detailed explanation of the scaled dot-product attention and the multi-head attention.

Position encoding is intended to enable the Transformer model to perceive the relationships between elements' positions and orders when processing EEG. In this experiment, we employed a commonly used relative position encoding method. Different position encoding methods will be considered in future work, as we have mentioned in the discussion section.

Thank you for your suggestion. We will consider real-time emotion recognition in our future work, and we have added an introduction to this in the discussion section.

Thank you for your suggestion. In future work, we will carefully consider user studies and subjective evaluations when collecting data.

Thank you again for your valuable feedback.

Reviewer 3 Report

Comments and Suggestions for Authors

The paper is interesting and eventually worth publishing but it needs quite an extensive revision.  

1)      First of all, the description of the method is rather superficial. There are a lot of separate diagrams but their description does not provide information on how they are linked together. The reader has to figure out on his own how the model is built. Definitely, a single diagram linking all the parts of the model/models proposed by the Authors should be provided.

2)      Second, it is impossible to replicate the results provided in the paper, since there are too little information on the data fed to the model and on the model/training parameters. For example it is unknown: i) what is the actual dimension of the input matrix fed to the model, ii) how many kernels in CNN where used, iii) what are the temporal and spatial inputs in Fig. 3 (both have the same dimension provided in green boxes), iv) how long the model was trained, v) which EEG channels were applied from both datasets, etc.

3)      Third, it is difficult to understand the model results. The paper is based on two datasets SEED and DEAP but there is only one set of results presented. I would presume that those are the results obtained after concatenating both sets but it is rather impossible since both sets have different number of channels.

4)      The Authors used ten-fold cross validation to validate the model results, without validating it on the external test set. Moreover, the validation and training sets are not orthogonal, both are based on the data from the same subjects. Both issues make the testing process not quite reliable. The question is how the model will behave with data from different subject?

5)      No information on the statistical significance of the differences in results obtained with each of the models is provided.

6)      The accuracies presented in Table 5 (last row) do not agree with the confusion matrixes.

7)      The references in Table 5 are not correct.

8)      Although the paper deals with DL methods, the SoA presented in Introduction is focused on classic ML approaches (only one DL paper is cited).

Author Response

Dear Reviewer,

Thank you for reviewing our paper and providing valuable feedback. We greatly appreciate your attention to our research and are willing to provide detailed responses and revisions to the questions you have raised.

Below are the corresponding revisions and responses:

1)Based on your suggestion, we described in Section 2.5 how Figure 2 (a), (b), and (c) are connected. Figure 3 depicts the overall architecture of the model we designed, employing the Transformer encoder.

2)Based on your suggestions, we have added the number of kernels used by CNN in Figure 3. In the dataset introduction section, we provided information on the channel numbers for the SEED dataset and the DEAP dataset, which are 62 and 32 respectively, and also included which channels were used in these two datasets. In Section 3.4, regarding the model input, we provided a more detailed explanation of the spatial and temporal arrangement of EEG.

3)We conducted separate experiments on the SEED and DEAP datasets. The SEED dataset classifies emotions as Positive-Neutral-Negative, while the DEAP dataset classifies emotions based on Arousal and Valence. Therefore, there are four dimensions: Positive-Neutral-Negative, Arousal-Valence, Arousal, and Valence. The experimental results are obtained separately for each of these four dimensions.

4)Thank you very much for raising this question. In our experiment, we use the same dataset for both training and validation, which may indeed result in the training and validation sets not being completely orthogonal. However, the SEED and DEAP datasets have different numbers of channels, and our model cannot be trained on one dataset and validated on the other. Furthermore, when conducting experiments on a dataset, we mix together the data from all subjects and then employ cross-validation to partition the dataset and evaluate the model.

5)Following your suggestion, we conducted t-tests on the results of the four models across different dimensions.

6)We have dealt with the problem, thank you for pointing it out.

7)We have dealt with the problem, thank you for pointing it out.

8)We have added references on deep learning in the introduction section.

Thank you again for your valuable feedback.

Reviewer 4 Report

Comments and Suggestions for Authors

This is an interesting paper describing a new deep learning method for computing emotions from EEG data using transformers and CNN. Using the temporal-spatial nature of EEG data for emotion prediction makes a lot of sense.

What I do not understand is how the two test datasets SEED and DEAP have been used for the verification of the proposed algorithm? Have they been combined into one dataset for validation, or have the averages been taken? How have the numbers in table 1 and figures 5 and 6 been computed?

It would also be nice to identify a scenario where the proposed algorithm is significantly better as previous work on the same data, as there is previous work using LSTM achieving similarly high accuracy. As the N of both datasets is quite small (15 to 20 people each), all of these approaches will overfit on the training dataset. A real proof would be to apply the model to newly collected data in the wild.

Comments on the Quality of English Language

The English is well understandable although not fully correct. There are many articles and pronouns missing, as is typical for English written by native Chinese speakers.

Author Response

Dear Reviewer,

Thank you for reviewing our paper and providing valuable feedback. We greatly appreciate your attention to our research and are willing to provide detailed responses to the questions you have raised.

Below are the corresponding revisions and responses:

In your review, you raised questions about how to validate our proposed algorithms using SEED and DEAP datasets. We conducted separate experiments on the SEED and DEAP datasets. The SEED dataset classifies emotions as Positive-Neutral-Negative, while the DEAP dataset classifies emotions based on Arousal and Valence. Therefore, there are four dimensions: Positive-Neutral-Negative, Arousal-Valence, Arousal, and Valence. The experimental results are obtained separately for each of these four dimensions.

In our study, we aimed to come up with a novel approach to SEED and DEAP datasets, and we have taken some steps to address the overfitting problem when designing our experiments. Despite the relatively small size of the dataset, we have taken a series of precautions, such as using cross-validation and regularization techniques to mitigate the effects of overfitting.In view of your suggestions, we will further consider how to prove the superiority of our proposed algorithm. We plan to apply the model to the newly collected data in future work to evaluate its performance . This will give us a more convincing validation and ensure the feasibility of our approach.

Thank you again for your valuable feedback.

Reviewer 5 Report

Comments and Suggestions for Authors

The authors of the manuscript titled “Emotion classification based on Transformer and CNN for EEG spatial-temporal feature learning” propose a novel model based on Transformer and Convolutional Neural Networks (TCNN) for EEG spatial-temporal (EEG ST) feature learning for automatic emotion classification. The research topic is interesting, and the authors have done a commendable job in writing their research manuscript.

I have some feedback to share with the authors:

·      The manuscript contains acronyms that need to be addressed (e.g., RNN, LSTM, etc.).

·      I would suggest carefully checking the statement in lines 47-48.

·      I would recommend careful verification of the statement regarding 0 to 75 Hz in line 103.

·      In lines 113-115, I suggest using “Epoching” instead of “segmenting”. Also, it would be beneficial to verify if this corresponds to the scientific correctness.

·      I recommend referencing the content in lines 115-116.

·      In line 120, I suggest using “implement” instead of “trained”.

·      I would suggest numbering each block in Figure 1 and providing a more exhaustive caption.

·      In Figure 3, I recommend adding the layer size using CNN theory.

·      Providing more information about data pre-processing and dataset organization and size used to train, test, and validate CNNs would be beneficial.

·      Including a table comparing CNN architectures would enhance the manuscript.

·      Figure 4 is not clear. It appears that each bar represents one value, such as mean and standard deviation. Perhaps a table would be easier to visualize.

·      Using ROC curves to visualize the learning process of the CNN used would be beneficial.

·      I recommend considering the following paper for reference: “Finger pinching and imagination classification: A fusion of CNN architectures for IoMT-enabled BCI applications”.

·      Table 5 has some issues with references that need to be addressed.

·      Improving the limitation and future directions section would enhance the manuscript.

·      Improving the discussion by contrasting the results of the literature with what the authors have achieved would be beneficial.

Comments on the Quality of English Language

The language of this manuscript needs a thorough revision.

Author Response

Dear Reviewer,

Thank you for reviewing our paper and providing valuable feedback. We greatly appreciate your attention to our research and are willing to provide detailed responses and revisions to the questions you have raised.

Below are the corresponding revisions and responses:

The issue with acronyms has been addressed.

We changed "nervous system disorders" to "neurological disorders" at lines 47-48.

At line 103, we changed "Furthermore, a bandpass filter ranging from 0 to 75Hz was applied for signal filtering."  to "Furthermore, filtering was performed using a bandpass filter with a frequency range of 0 to 75 Hz. "

At lines 113-115, upon reviewing the literature, we believe that using 'segment' here would be more appropriate.

At lines 115-116, references have been added.

At line 120, we replaced 'trained' with 'implemented'.

We numbered each block in Figure 1 and provided more detailed explanations.

In Figure 3, we added the sizes of the CNN layers.

We added a table comparing the statistical results of differences in outcomes obtained by each model.

We carefully consulted the paper titled 'Finger pinching and imagination classification: A fusion of CNN architectures for IoMT-enabled BCI applications'.

The issue with incorrect reference citations in Table 5 has been resolved.

Thank you again for your valuable feedback.

Round 2

Reviewer 2 Report

Comments and Suggestions for Authors

The authors considered my comments and suggestions 

Comments on the Quality of English Language

A final proofread would be useful 

Reviewer 4 Report

Comments and Suggestions for Authors

this looks good to me now